# LHCSR3-Type NPQ Prevents Photoinhibition and Slowed Growth under Fluctuating Light in *Chlamydomonas reinhardtii*

**DOI:** 10.3390/plants9111604

**Published:** 2020-11-18

**Authors:** Thomas Roach

**Affiliations:** Department of Botany and Centre for Molecular Biosciences Innsbruck, University of Innsbruck, Sternwartestraße 15, 6020 Innsbruck, Austria; thomas.roach@uibk.ac.at; Tel.: +43-51250751030; Fax: +43-51250751099

**Keywords:** fluctuating light, photoinhibition, LHCSR, NPQ, *Chlamydomonas reinhardtii*, stress, state transitions

## Abstract

Natural light intensities can rise several orders of magnitude over subsecond time spans, posing a major challenge for photosynthesis. Fluctuating light tolerance in the green alga *Chlamydomonas reinhardtii* requires alternative electron pathways, but the role of nonphotochemical quenching (NPQ) is not known. Here, fluctuating light (10 min actinic light followed by 10 min darkness) led to significant increase in NPQ/qE-related proteins, LHCSR1 and LHCSR3, relative to constant light of the same subsaturating or saturating intensity. Elevated levels of LHCSR1/3 increased the ability of cells to safely dissipate excess light energy to heat (i.e., qE-type NPQ) during dark to light transition, as measured with chlorophyll fluorescence. The low qE phenotype of the *npq4* mutant, which is unable to produce LHCSR3, was abolished under fluctuating light, showing that LHCSR1 alone enables very high levels of qE. Photosystem (PS) levels were also affected by light treatments; constant light led to lower PsbA levels and *F*_v_/*F*_m_ values, while fluctuating light led to lower PsaA and maximum P700^+^ levels, indicating that constant and fluctuating light induced PSII and PSI photoinhibition, respectively. Under fluctuating light, *npq4* suffered more PSI photoinhibition and significantly slower growth rates than parental wild type, whereas *npq1* and *npq2* mutants affected in xanthophyll carotenoid compositions had identical growth under fluctuating and constant light. Overall, LHCSR3 rather than total qE capacity or zeaxanthin is shown to be important in *C. reinhardtii* in tolerating fluctuating light, potentially via preventing PSI photoinhibition.

## 1. Introduction

Photosynthetic bacteria, algae, and plants are able to cope with rapid fluctuations in light intensity, although this requires significantly more regulation of photosynthesis than light of constant intensity [1,2]. Rates of CO_2_ assimilation are limited and cannot always keep pace with rapid changes in light intensity that happen, e.g., as clouds pass across the sun or in sun spots under canopies. Nonphotochemical quenching (NPQ) is a general term for mechanisms that regulate how much light energy is available for photosynthesis [3]. NPQ is considered important in preventing elevated formation of reactive oxygen species (ROS) that otherwise could occur during excess light absorption [4,5]. The so-called qE component of NPQ, which reduces quantum yields of chlorophyll fluorescence in the light-harvesting antennae, is the dominant thermal dissipation pathway driven by pH changes in the thylakoid lumen [6,7,8]. In contrast to the constitutively high qE capacity found in many land plants, the fresh water alga *Chlamydomonas reinhardtii* has a flexible qE capacity that responds to the environment [9]. Under low light or high light when supplemented with organic carbon (both conditions that do not lead to excess absorption of light energy), the qE capacity of *C. reinhardtii* is minimal. Only under energetic imbalances, resulting from low CO_2_ availability and excess light or exposure or photo-oxidative stress, qE capacity is increased to safely mitigate excess-absorbed light energy [10,11,12].

In *C. reinhardtii*, thermal dissipation of excess light energy via qE is intricately linked to Light-Harvesting Complex (LHC)-like Stress-Related (LHCSR) thylakoid membrane proteins, LHCSR1 and LHCSR3 [13], which have been lost in plants through evolution. LHCSR1-mediated quenching was shown to occur in LHCII [14,15], or energy transfer via LHCII to PSI [16], while LHCSR3-mediated qE occurs in PSII-LHCII-LHCSR3 supercomplexes [17,18] or in LHCII-LHCSR3 associated to PSI [14]. The pH-regulated xanthophyll cycle (e.g., de-epoxidation of violaxanthin to zeaxanthin) is a ubiquitous response of photosynthetic organisms to high light. Zeaxanthin in proximity to photosynthetic complexes is involved in the qE of plants and some algae, but is not required for high qE in *C. reinhardtii* [19,20,21]. Further to its role in qE, zeaxanthin has several other stress-associated roles (e.g., an antioxidant [5,21]). Tobacco mutants with an altered xanthophyll cycle and accelerated relaxation of qE had elevated photosynthetic efficiency in the field under naturally fluctuating light conditions, relative to WT, which was attributed to prevention of lost photosynthesis by excess dissipation during a decrease in light intensity [22]. Increasing photosynthetic efficiency via tweaking NPQ, and specifically qE, could be one way to elevate growth rates of lipid-rich algae, which are considered one of the most promising future sources of renewable biofuels [23].

Previously, tolerance to fluctuating light in *C. reinhardtii* has focused on cyclic electron flow via PGR5 or PGRL1 proteins, and the importance of maintaining the donor side of PSI oxidized via flavodiirons to prevent PSI photoinhibition [24,25]. In the diatom *Phaeodactylum tricornutum,* qE-type NPQ is part of the fluctuating light acclimation [26], but so far, the role of qE in fluctuating light tolerance in *C. reinhardtii* has not been elucidated. Here, I investigated how NPQ mechanisms may protect against fluctuating light, by comparing the growth and photosynthetic response of the *npq4* mutant deficient in LHCSR3 to wild type (WT) and the *npq1* and *npq2* mutants with disrupted xanthophyll compositions.

## 2. Results

### 2.1. Light Fluctuations, but Not Constant Light, Slowed npq4 Growth

Colony density of *npq4* was visually much less than WT-4A when grown under repeated light fluctuations of 10 min exposure and 10 min darkness (Figure 1). This agreed with increases in fresh weight of WT-4A being 2.5-fold and 6-fold higher than *npq4* when fluctuating light intensities were 100 and 500 µmoL quanta m^−2^ s^−1^, respectively, after 6–8 day growth (Appendix A). Colony growth rates of the two genotypes were very similar under constant saturating (500 µmoL quanta m^−2^ s^−1^) and subsaturating (100 µmoL quanta m^−2^ s^−1^) light intensities (Figure 1 and Appendix A) and also when darkness was replaced with low light intensity (50 µmoL quanta m^−2^ s^−1^; Appendix A).

### 2.2. Light Fluctuations Induced Accumulation of LHCSR1, LHCSR3, and qE Capacity

Levels of LHCSR proteins, and therefore qE capacity (Figure 2), were clearly different under fluctuating and constant light. The level of LHCSR3, in WT, and LHCSR1, in WT and *npq4*, increased several folds under fluctuating light (Figure 3). This agreed with the higher Y(NPQ) values relative to those found in cells grown in constant light at the same intensity (Figure 2B). Even at 100 µmoL quanta m^−2^ s^−1^ fluctuating light, Y(NPQ) of *npq4* increased so much that it equaled WT-4A (Figure 1 and Figure 2B). There was no consistent influence of initial culture dilution on Y(NPQ) values.

### 2.3. Enhanced PSII and PSI Photoinhibition Occurred in npq4 Under Constant and Fluctuating Light, Respectively

Values of *F*_v_/*F*_m_, here used as a proxy for PSII photoinhibition, were generally higher in WT-4A than *npq4* under all light treatments. Notably, at 500 µmoL quanta m^−2^ s^−1^, *F*_v_/*F*_m_ values of *npq4* were much lower under constant than fluctuating light (Figure 1 and Figure 2A). This agreed with the lowest PsbA levels (PSII reaction center) found in *npq4* under constant light and much higher levels in cells grown under fluctuating light (Figure 3). However, under fluctuating light WT-4A and *npq4* had lower levels of PsaA (PSI reaction center), relative to constant light-treated cells (Figure 3 and Appendix A). This agrees with significantly altered maximum P700 redox changes (*P*_m_), here used as a proxy for PSI photoinhibition. Lowered *P*_m_ values were found in both genotypes under fluctuating light, with *npq4* significantly lower than WT-4A (Figure 4).

### 2.4. Xanthophyll Composition Did Not Affect Tolerance to Fluctuating Light

Under fluctuating light, *npq4* contained significantly more (*p* = 0.02) antheraxanthin and slightly more zeaxanthin, and significantly less (*p* < 0.01) violaxanthin, showing higher activation of the xanthophyll cycle than in WT-4A (Figure 5A). While it is known that de-epoxidized xanthophylls do not play a significant contribution to qE in *C. reinhardtii* [19], the xanthophyll cycle is still tightly coupled to high light exposure [27]. To investigate if differences in xanthophyll composition also influenced tolerance to fluctuating light, the NPQ mutants *npq1* (no zeaxanthin) and *npq2* (only zeaxanthin, and neither violaxanthin nor neoxanthin [27]) were included in the study. No differences in colony growth between *npq1* and *npq2* were observed under constant or fluctuating light (Figure 5B), but similar to the other genotypes, fluctuating light increased qE capacity and led to higher *F*_v_/*F*_m_ values relative to cells under constant light of the same intensity (Figure 5B).

## 3. Discussion

Regulation of electron transport is of general importance for efficient photosynthesis and to prevent photoinhibition, especially under fluctuating light [1,28]. NPQ influences electron transport rates [20] and in plants has been shown to have both positive and negative impacts on plant growth, under constant and fluctuating light [22,29]. Growth of *npq4* was only mildly affected under constant light (Figure 1), which is in agreement with other studies using LHCSR-deficient mutants at similar light intensities, at least under ambient oxygen tensions [10,30]. *C. reinhardtii* responded to fluctuating light by increasing levels of LHCSR1 and LHCSR3 (Figure 3), inferring that these qE proteins are important in the dynamic regulation of photosynthesis under fluctuating light. Surprisingly, Y(NPQ) values of *npq4* were as high as WT under fluctuating light (Figure 2B), which must have been strictly mediated by LHCSR1, due to the absence of LHCSR3 in this mutant [13].

LHCSR-mediated qE is dependent on pH [20] and becomes active in response to protonation of luminal-exposed part of the protein [7,8] that naturally occurs under high light. The xanthophyll cycle is another high light-associated process regulated by pH, but has limited contribution to qE of *C. reinhardtii* and other Chlorophyta alga [19,31]. Absence of an active xanthophyll cycle in the *npq1* and *npq2* mutants did not affect tolerance to fluctuating light (Figure 5B). Therefore, higher levels of de-epoxidized xanthophylls in *npq4* under fluctuating light compared to WT (Figure 5A) is unlikely to have contributed to any change in fluctuating light tolerance. Nonetheless, a greater shift in the de-epoxidation ratio of *npq4* is indicative of a lower luminal pH than occurred in WT. Since the level of LHCSR-mediated qE is directly associated to pH [20], low pH values may provide an explanation of the particularly high Y(NPQ) values in *npq4* in response to fluctuating light (Figure 2B). We can also be confident from the behavior of *npq4* that LHCSR1 is able to induce a large, fast, and reversible pH-dependent quenching of LHCII [15] in response to fluctuating light.

Under fluctuating light, a lower luminal pH and elevated electric field across the thylakoid membrane (Δ*Ψ*) increases incidences of charge recombination, ROS production, and PSII photoinhibition in *Arabidopsis thaliana* [32]. In contrast, here, *F*_v_/*F*_m_ values of *npq4* were higher under fluctuating light, relative to constant light (Figure 2A), indicating that fluctuating light did not induce PSII photoinhibition as much as under constant light. Indeed, higher levels of PsbA, the D1 reaction center of PSII, were likewise found under fluctuating than under constant light (Figure 3 and Appendix A). PSII photoinhibition directly impacts linear electron flow, potentially influencing PSI photoinhibition [28].

So far, it has been shown that PGR5/PGRL1-mediated cyclic electron flow contributes to fluctuating light tolerance in photosynthetic organisms [25,33], but in *C. reinhardtii*, at least, the role of flavodiirons in preventing acceptor side limitation of PSI is more critical. In the absence of flavodiirons, fluctuating light leads to PSI photoinhibition [24,25]. This highlights that PSI can be rendered vulnerable under fluctuating light. Under constant light, enhanced PSI photoinhibition has been observed in *npq4*, but only under elevated oxygen tensions [10], or in cells also deficient in PGRL1-mediated cyclic electron flow [34,35]. Western blotting (Figure 3) and P700 absorbance measurements (Figure 4) revealed that repeated 10 min light–dark treatments led to decrease in PSI levels, particularly in *npq4*. Photodamage of PSI can be exacerbated by high rates of electron flow from PSII [36]. Therefore, a lack of LHCSR3 and potentially low qE in *npq4* could contribute to enhanced PSI photoinhibition. However, in fluctuating light, under which PSI photoinhibition of *npq4* occurred, Y(NPQ) values were the same as in WT (Figure 2B), indicating that the influence of qE on linear electron flow would have been equal in both genotypes.

Another explanation to enhanced PSI photoinhibition in *npq4* under fluctuating light would concern state transitions. This phosphorylation-mediated NPQ mechanism [37] is particularly active in *C. reinhardtii* during the first few minutes of light-to-dark and dark-to-light acclimation and is affected in *npq4* [38]. High light-treated cells are in state 1, but when subjected to darkness they transit to state 2 due to chlororespiration-induced phosphorylation of specific components of the antenna, including LHCII and LHCSR3 [39,40]. During transition to state 2, the majority of LHCII migrates energy transfer to PSI [41], and LHCSR3 also migrates as part of the LHCSR3-LHCII antenna of PSI [40]. Since LHCSR3 can quench LHCII associated with PSI [14], it is possible that LHCSR3 directly protects PSI from photodamage during fluctuating light, when cells are repeatedly exposed to high light in state 2 [38]. Unlike LHCSR3, LHCSR1 is not phosphorylated [34], and therefore unlikely to be in the mobile LHCII fraction during transition to state II. While the large increase in LHCSR1 levels (Figure 3) suggests that LHCSR1-mediated qE is important under fluctuating light, without LHCSR3, as in *npq4*, LHCSR1 alone could not prevent PSI photoinhibition (Figure 4 and Figure 5). High LHCSR1 levels may have even promoted PSI photoinhibition, since LHCSR1 has been reported to quench LHCII via PSI [16]. Finally, *npq4* was not growth-sensitive to repeated 10-fold increases in light intensity (i.e., when darkness was replaced by low light; Appendix A). Therefore, deactivation of the Calvin–Benson cycle during 10 min of darkness [42] was likely important to reveal the sensitivity of *npq4* to fluctuating light.

## 4. Conclusions

Overall, LHCSR3-mediated NPQ rather than overall qE capacity is important in tolerating fluctuating light that involves darkness. In agreement with previous studies, PSI showed vulnerability to fluctuating light, and with the use of *npq4,* it was possible to show that LHCSR3 can protect PSI, a role that LHCSR1 seems not to be able to fulfill. In contrast to the efficient repair cycle of PSII, photoinhibition of PSI is much more costly due to very slow PSI repair rates [43]. Therefore, enhanced photoinhibition of PSI due to absence of LHCSR3 can explain the growth phenotype of *npq4* under fluctuating light. Interactions between LHCSR1/3 and LHCII with PSI in energy dissipation in response to dynamic light exposure require further elucidation.

## 5. Material and Methods

### 5.1. C. reinhardtii Strains and Growth Conditions

The *C. reinhardtii* LHCSR3-deficient strain *npq4* (CC-4614) and its parental WT-4A (CC-4051) were purchased from the Chlamydomonas Centre (www.chlamycollection.org). Liquid cultures were initiated in photoautotrophic media (THP), which was identical to TAP except acetic acid was replaced by HCl. Cultures were adjusted to 5.0, 1.0, and 0.5 µg chlorophyll mL^−1^ or 12,500, 2500, and 1250 cells (from here-on referred to as 1:1, 1:5, and 1:10 dilutions, respectively), in the 10 µL starting culture pipetted onto THP media containing 1.5% agar. Each Petri dish contained four replicates of each strain placed in an alternating order, with different plates hosting each dilution. The Petri dish lids were placed on, but not sealed with any film or tape to allow gas exchange, before placing in a growth chamber (Percival, PGC-6HO) at 25 °C under subsaturating or saturating light intensity (100 or 500 µmoL photons m^−2^ s^−1^, respectively) with a 24 h time span composed of either constant 12 h illumination or repeated light fluctuations of 10 min illumination and 10 min darkness (see Appendix A for a profile of the light intensity during one light fluctuation cycle). This light cycle was chosen to enable activation and relaxation of LHCSR3-associated NPQ processes [35]. For comparing growth rates, all colonies of the same genotype were carefully scraped, using a flat-ended spatula, together from the agar and the fresh weight was divided by the time colonies had been growing, which were 6, 7, and 8 days for 1:1, 1:5, and 1:10 dilutions, respectively. Additional cultures initiated from 1:1, 1:5, 1:10, and 1:25 dilutions were treated with fluctuating light as above, except that darkness was replaced by low light (50 µmoL photons m^−2^ s^−1^) to prevent deactivation of the Calvin–Benson cycle. All dilutions under this treatment were weighed after 7 days growth.

### 5.2. Chlorophyll Fluorescence

At the end of the 12 h constant light treatment on day 6, 7, and 8 for 1:1, 1:5, and 1:10 dilutions, respectively, cultures from both light treatments were moved to darkness for 1 h. After removing the Petri dish lid, chlorophyll fluorescence during a 600 ms saturating pulse was measured with a Maxi Imaging PAM M-series (WALZ). Minimum (*F*) and maximum fluorescence (*F*_m_) was used to calculate maximum PSII quantum yields (*F*_v_/*F*_m_) via (*F*_m_−*F*)/*F*_m_. Thereafter, Y(NPQ), as an indicator of the fraction of light energy dissipated to heat via qE, was measured after 30 s at 396 µmoL photons m^−2^ s^−1^ with a subsequent saturating pulse and calculated by (*F*/*F*_m_′)−(*F*/*F*_m_), with *F*_m_ and *F*_m_′ measured before and during light, respectively. Immediately after, cultures were frozen in liquid nitrogen.

### 5.3. Western Blotting of Proteins

For detecting specific proteins via Western blotting, the frozen samples were thawed and extracted in 50–150 µL (depending upon culture amount) of 2% SDS in 50 mM TRIS-HCl, pH 6.8, containing protease inhibitor cocktail (Roche). After centrifugation for 10 min at 4 °C and 16,000 × *g*, supernatants were measured for protein content using the BCA assay for loading on 12% acrylamide gels at equal protein level (1 µg for PsbA and LHCSR3 and 10 µg for PsaA and LHCSR1; see Appendix A for showing blots were below saturation point), before running for 1.5 h at 150 V, semi-dry transfer to nitrocellulose membranes, and incubation with antibodies (Agrisera), according to [10].

### 5.4. HPLC of Photosynthetic Pigments

Photosynthetic pigments were measured by HPLC in cultures grown from 1:10 dilution as above, except cells grown across the whole agar plate rather than from individual 10 µL spots (*n* = 3 plates/genotype). Pigments from ca. 2 mg of lyophilized cultures were extracted in 0.5 mL of ice-cold acetone and measured by absorbance at 440 nm, after separation with an Agilent 1100 HPLC system equipped with a LiChrospher 100 RP-18 column (125 mm × 4 mm, 5 µm), according to [11].

### 5.5. Maximum Photo-Oxidizable P700 Level

Maximum photo-oxidizable P700 levels (*P*_m_) were measured during a 200 ms saturating pulse using a DUAL-PAM (Walz), of cultures grown as for pigment measurements (see Section 2.4). Cells were scraped off the agar and suspended in THP liquid media (see Section 2.1) at equal total chlorophyll concentration of 30 µg mL^−1^, and measured according to Roach, Na, Stöggl, and Krieger-Liszkay [10].

### 5.6. Statistics

For chlorophyll fluorescence measurements, four individual colonies were used as replicates for each culture dilution. Data were analyzed by two-way ANOVA in SPSS Statistics 25 (IBM) to reveal *p*-values between WT and *npq4*, considering all dilutions collectively (*n* = 12 colonies) for each light treatment. For comparing colony growth, average fold-difference in fresh weight accumulation for the three culture dilutions (1:1, 1:5, and 1:10) was compared with a Students *t*-test under each light treatment. Significant differences were considered when *p* < 0.05.

## Figures and Tables

**Figure 1 plants-09-01604-f001:**
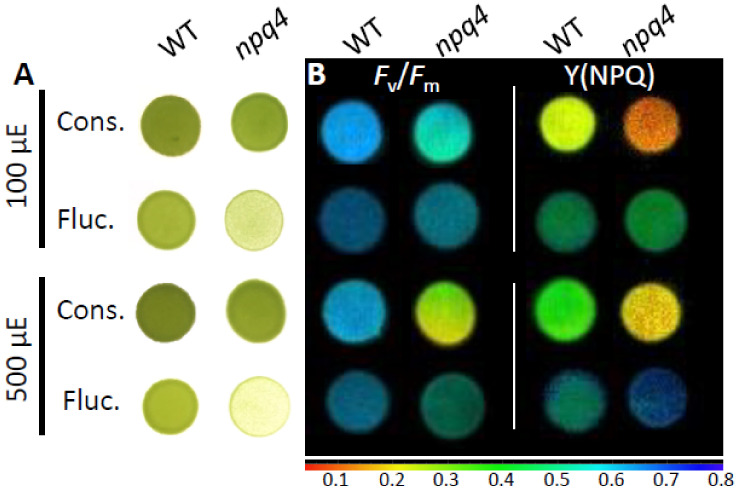
Phenotype of Light-Harvesting Complex (LHC)-like Stress-Related 1 (LHCSR3)-deficient *nqq4* relative to wild type (WT-4A) under photoautotrophic conditions on agar and either diurnal constant (12/12 h, on/off) or fluctuating (10/10 min, on/off) light treatments. Representative images of (**A**) colonies and (**B**) *F*_v_/*F*_m_ and Y (non-photochemical quenching (NPQ)) after 8 day growth (starting dilution 1:10, see Materials and Methods) under diurnal/constant (Cons.) or fluctuating (Fluc.) light of 100 (upper) or 500 (lower) µmoL photons m^−2^ s^−1^. Chlorophyll fluorescence parameters are given on a false-color scale shown below.

**Figure 2 plants-09-01604-f002:**
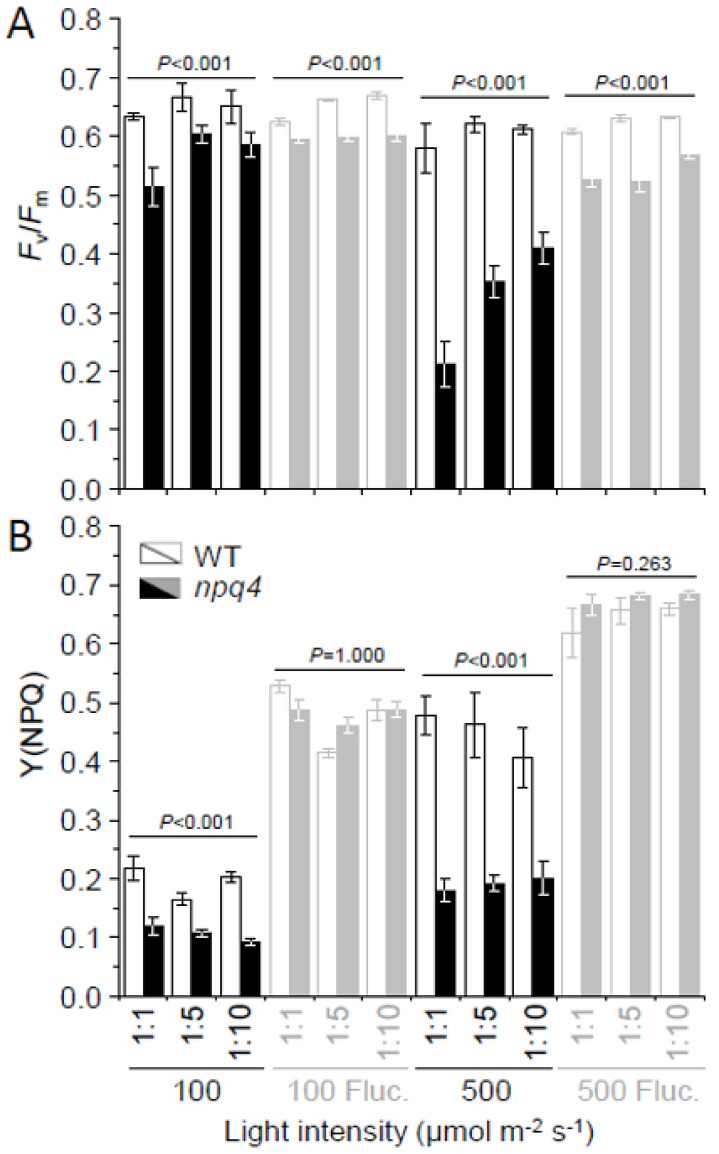
Effect of either diurnal constant or fluctuating light treatments on (**A**) *F*_v_/*F*_m_ and (**B**) Y(NPQ) values of photoautotrophic agar-grown cells. Colonies were cultured under 100 or 500 µmoL photons m^−2^ s^−1^, of either diurnal constant (12/12 h on/off; black filled or black-outlined bars) or fluctuating light (10/10 min on/off, gray filled or gray-outlined bars). Wild type (WT-4A; open bars) and LHCSR3-deficient *npq4* (closed bars) are shown side by side (*n* = 4 replicate colonies ±SD). The *p*-values from two-way ANOVA are shown for comparisons of *npq4* to WT-4A for each light treatment when considering all differing initial culture dilutions together (*n* = 12 colonies).

**Figure 3 plants-09-01604-f003:**
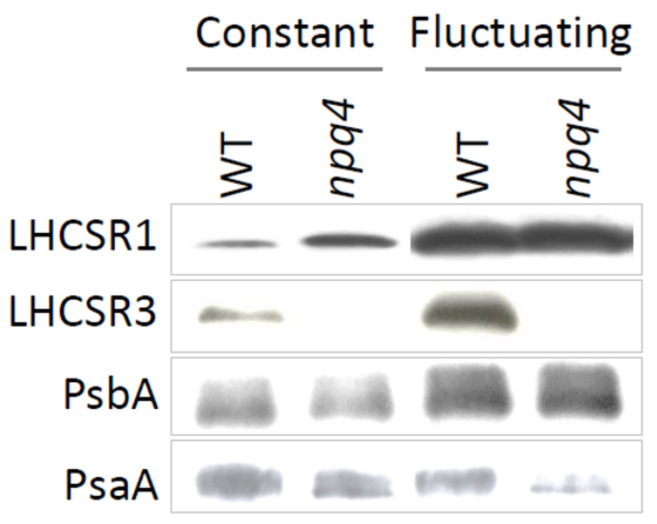
Effect of 500 µmoL photons m^−2^ s^−1^ fluctuating (10/10 min on/off) or constant (12/12 h on/off) light treatment on qE proteins and photosystem reaction center levels of photoautotrophic agar-grown cells. Levels of LHCSR1, LHCSR3, PsbA, and PsaA are shown, via Western blotting, in LHCSR3-deficient *npq4* and wild type (WT-4A) grown under constant (left) or fluctuating light (right). In this Figure, exactly the same sample from each treatment was loaded for all four blots. Loading controls for saturation are shown in Appendix A.

**Figure 4 plants-09-01604-f004:**
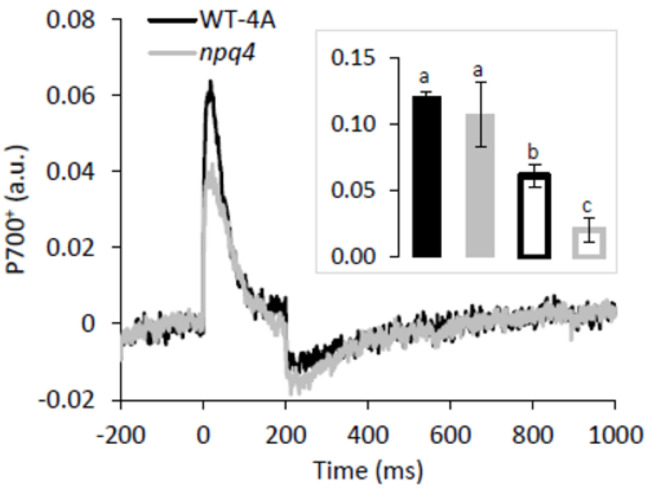
Effect of 500 µmoL photons m^−2^ s^−1^ fluctuating (10/10 min on/off) or constant (12/12 h on/off) light treatment on maximum light-induced P700^+^ values of photoautotrophic agar-grown cells. Wild type (WT-4A; black) and LHCSR3-deficient *npq4* (gray) agar-grown cells were suspended in photoautotrophic media to a chlorophyll concentration of 30 µg mL^−1^, and P700^+^ was recorded during a 200 ms saturating pulse, starting at 0 ms, with typical traces shown after averaging the three technical replicates of fluctuating light-treated cells. The inset shows average *P*_m_ values for *npq4* and WT under constant (filled) or fluctuating light (open), *n* = 4 replicate cultures ± SD, with different letters denoting significant difference, *p* < 0.05.

**Figure 5 plants-09-01604-f005:**
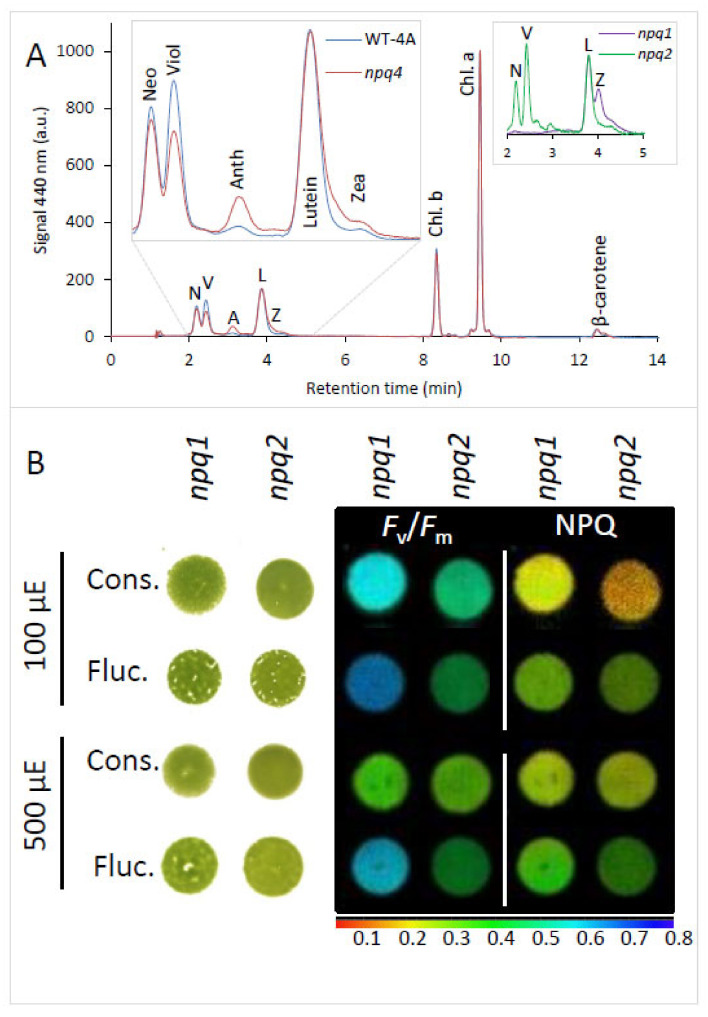
Xanthophyll levels under fluctuating light and lack of influence of xanthophyll compositions on tolerance to fluctuating light. (**A**) Typical HPLC chromatograms showing the relative amounts of neoxanthin (Neo), violaxanthin (viol), antheraxanthin (anth), Lutein, and zeaxanthin (zea) of wild type (WT-4A) and *npq4* under 500 µmoL photons m^−2^ s^−1^ fluctuating light. The insets are an expansion of 2–5 min, with left WT-4A (blue) and *npq4* (red) and right zeaxanthin epoxidase-deficient (*npq1*; purple) and violaxanthin de-epoxidase-deficient (*npq2*; green) mutants. (**B**) Colonies of *npq1* and *npq2* are shown side by side after growth under constant (cons.) or fluctuating (fluc.) light at 100 (upper) or 500 (lower) µmoL photons m^−2^ s^−1^. Chlorophyll fluorescence parameters, *F*_v_/*F*_m_ and qE-NPQ are shown on a false-color scale (below).

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
