# Peer review of "LHCSR3-Type NPQ Prevents Photoinhibition and Slowed Growth under Fluctuating Light in Chlamydomonas reinhardtii"

_plants, 2020, doi:10.3390/plants9111604_

Round 1
Reviewer 1 Report
This paper evaluates the phenotype of npq knockout strains of Chlamydomonas reinhardti under fluctuating light condition. It is well written and the scientific conclusions sound.
I only have a concern concerning the western blot (fig3.) since no loading control nor check of the transfer (e.g. ponceau stain) is provided. After the materials and methods section, I assume that 4 membranes are shown, one per antibody. I would suggest to used black frames to visually show it (see fig.7 ).
Minor:
Figure caption are missing, please provide them
Supplementary Fig. 2= PsaB shall be changed for PsbA
Author Response
I would like to thank the reviewer for their time and improvements suggested for the manuscript.
Reviewer: I only have a concern concerning the western blot (fig3.) since no loading control nor check of the transfer (e.g. ponceau stain) is provided. After the materials and methods section, I assume that 4 membranes are shown, one per antibody. I would suggest to used black frames to visually show it (see fig.7 ).
Response: Unfortunately, I do not have an image of Ponceau, although this was of checked, and admittedly this would be ideal to show. At least, exactly the same sample was used for each blot in this Figure. The legend was changed in Figure 3 to state this. Black borders were added to the edge of each blot in Fig 3 and Supplementary Figure 2.
Reviewer: Minor:
Figure caption are missing, please provide them
Response: They are after the references
Supplementary Fig. 2= PsaB shall be changed for PsbA
Response: changed
Reviewer 2 Report
Review result of Manuscript ID: plants-985540
Using the npq mutants, the correlative studies are carried out to identify the role of LHCSR in the regulation of NPQ in Chlamydomonas in the acclimation to fluctuating light. Western blot and physiological parameters (NPQ and Fv/Fm together with P700 fluorescence) provide evidence showing a critical role of LHCSR3 in the protection of PSI against photoinhibition.
The previous studies on the relationships between xanthophyll cycle and light acclimation in Chlamydomonas as well as Arabidopsis could be provided in the Introduction.
By the use of npq1 and npq2 mutants, authors claimed that xanthophyll cycle is not involved in the tolerance to fluctuating light. However, the xanthophyll cycle is affected in npq4 mutants. Please discuss it in the Discussion.
How to measure the FW of Chlamydomonas in supplementary Figure 1A? Although the Figure legend describes ‘(see methods)’, it is not found in the Methods for the determination of fresh weight. Is it collecting from agar plate and then weighs on the scale? How to collect the cells from agar plate?
It is suggested in supplementary Figure 3 that it could be better for drawing the plot of fluctuating light intensity with 10 min light-10 min darkness cycle from 0 min to 20 min.
Author Response
I would like to thank the reviewer for their time and improvements suggested for the manuscript.
Reviewer: The previous studies on the relationships between xanthophyll cycle and light acclimation in Chlamydomonas as well as Arabidopsis could be provided in the Introduction.
Response: The text from Line 23 was modified to: "The pH-regulated xanthophyll cycles (e.g. de-epoxidation of violaxanthin to zeaxanthin) is a ubiquitous response of photosynthetic organisms to high light. Zeaxanthin in proximity to photosynthetic complexes is involved in the qE of plants and some algae, but is not required for high qE in C. reinhardtii [19-21]. Further to its role in qE, zeaxanthin has several other stress-associated roles (e.g. an antioxidant [5, 21])."
Reviewer: By the use of npq1 and npq2 mutants, authors claimed that xanthophyll cycle is not involved in the tolerance to fluctuating light. However, the xanthophyll cycle is affected in npq4 mutants. Please discuss it in the Discussion.
Response: The text from Line 93 was changed to: "The xanthophyll cycle is another high light-associated process regulated by pH, but has limited contribution to qE of C. reinhardtii and other Chlorophyta alga [19,31]. Absence of an active xanthophyll cycle did not affect tolerance to fluctuating light (Fig. 5B). Therefore, higher levels of de-epoxidised xanthophylls in npq4 under fluctuating light (Fig. 5A) is unlikely to have contributed to any change in fluctuating light tolerance. Nonetheless, a greater shift in the de-epoxidation ratio of npq4 is indicative of a lower luminal pH than occurred in WT. Since the level of LHCSR-mediated qE is directly associated to pH [20], low pH values may provide an explanation of the particularly high Y(NPQ) values in npq4 in response to fluctuating light (Fig. 2B).
Reviewer: How to measure the FW of Chlamydomonas in supplementary Figure 1A? Although the Figure legend describes ‘(see methods)’, it is not found in the Methods for the determination of fresh weight. Is it collecting from agar plate and then weighs on the scale? How to collect the cells from agar plate?
Response: The text from Line 168 was changed to: "For comparing growth rates, all colonies of the same genotype were carefully scraped, using a flat-ended spatula, together from the agar and the fresh weight was divided by the time colonies had been growing, which were 6, 7 and 8 days for 1:1, 1:5 and 1:10 dilutions, respectively."
Reviewer: It is suggested in supplementary Figure 3 that it could be better for drawing the plot of fluctuating light intensity with 10 min light-10 min darkness cycle from 0 min to 20 min.
Response: The X-axis has been changed to start from 0 min.
Round 2
Reviewer 1 Report
The author addressed my concern, therefore I agree with the publication of the paper in its present form.